

# The antioxidant system response to drought-stressed *Diospyros lotus* treated with exogenous melatonin

Peng Zhang, Yi Hu[1], Ruijin Zhou, Xiaona Zhang[1,2], Huiling Hu and Dongmei Lang

School of Horticulture and Landscape Architecture, Henan Institute of Science and Technology, Xinxiang, China
Henan Province Engineering Research Center of Horticultural Plant Resource Utilization and Germplasm Enhancement, Xinxiang, China

Corresponding authors
Huiling Hu, hu-huiling@163.com
Dongmei Lang,
langdongmei1990@126.com

## ABSTRACT

Drought is one of the major abiotic stresses adversely impacting the growth of persimmon, which is a widely cultivated traditional fruit tree in North China. Melatonin is a bio-stimulator involved in mediating plant responses to drought. The role of exogenous melatonin application in the drought tolerance of *Diospyros lotus* was examined under drought stress with different doses of melatonin (0, 10, 50, and 100 $\mu$M). Exogenous melatonin application significantly mitigated the adverse effects of drought stress on chlorophyll fluorescence, lipid peroxidation, reactive oxygen species (ROS) accumulation and nitric oxide (NO) content. The 100-$\mu$M melatonin application produced the most beneficial impacts against drought stress. The melatonin-enhanced tolerance could be attributed to improved antioxidant enzymes, reduced drought-induced ROS accumulation, and lipid peroxidation. Melatonin application activated major antioxidant enzymes such as superoxide dismutase, catalase, peroxidase, glutathione reductase, and ascorbate peroxidase. Interestingly, NO concentration was significantly higher in 10 and 50 $\mu$M melatonin treatments and lower in 100 $\mu$M melatonin treatment compared to the control. Moreover, exogenous melatonin application affected the mRNA transcript levels of several genes involved in ROS metabolism, including *DlRBOHA*, *DlSOD*, *DlCAT*, and *DlPOD*. Hence, the responses of *Diospyros lotus* to drought varied with different doses of melatonin. Our results provide a concrete insight into the effects of melatonin with varying doses in alleviating drought as well as a platform for its potential application in the related fields.

## INTRODUCTION

Drought, one of the abiotic stresses, significantly limits growth and production in fruit trees. The 2021 IPCC Climate Change Report projects more frequent and/or severe droughts across all continents due to climate change (*Seneviratne et al., 2021*). China might face a severe drought risk (*Xu et al., 2021*). For example, since 2020, drought has caused a direct economic loss of 1.41 billion Chinese currency (*Guo, Chen & Pan, 2021*). Moreover, the areas experiencing moderate to high drought risk have expanded,

particularly in Northeast China (*Song et al., 2021*). Chlorophyll fluoroscence promptly responds to drought and can directly detect dynamic changes, so it is frequently used to monitor drought stress levels (*Shin et al., 2021*). Elevated drought stress inhibits root elongation and plant biomass (*Cuneo et al., 2021*; *Verslues & Longkumer, 2022*; *Wilschut & van Kleunen, 2021*), diminishes net photosynthesis, and thereby reduces carbohydrate concentrations, fruit production, and even causes death (*Ahmad et al., 2021b*; *Rowland et al., 2021*; *Wang et al., 2018*). In plants, drought stress always produces excessive reactive oxygen species (ROS), including superoxide ($O_2^{\bullet-}$) and hydrogen peroxide ($H_2O_2$) (*Zhang et al., 2019c*). The increased ROS production disturbs cells' redox balance, causing lipid peroxidation, ion leakage, and DNA-strand cleavage (*Rowland et al., 2021*). To combat the drought-induced ROS and oxidative stress, plants activate several protective mechanisms such as antioxidant enzyme production, including superoxide dismutase (SOD), catalase (CAT), peroxidase (POD), ascorbate peroxidase (APX), and glutathione reductase (GR) (*Laxa et al., 2019*; *Lukić et al., 2020*; *Rezayian, Niknam & Ebrahimzadeh, 2020*).

Persimmon (*Diospyros kaki*) is a traditional fruit tree in China with a production of 3.21 million tons, accounting for 75.18% of the world's persimmon production (*FAO, 2020*). Due to the fruit's high commercial value, persimmon is grown abundantly in the mountains of north China (*Griñán et al., 2019*). Drought is often the major environmental stress in this area; hence, drought tolerance of rootstocks could contribute to sustainable intensification of persimmon production. *Diospyros lotus* is the most common persimmon rootstock in North China (*Wei et al., 2015*). Although *D. lotus* is often considered a drought-tolerant plant, its drought adaption mechanism remains unclear.

The use of phytohormones to increase plants' tolerance to severe stress has recently drawn much attention (*Ciura & Kruk, 2018*; *Jogawat et al., 2021*; *Ullah et al., 2018*). Melatonin (N-acetyl-5-methoxytryptamine) is present ubiquitously in higher plants (*Arnao & Hernández-Ruiz, 2019*), and the endogenous melatonin concentration varies in different species, different varieties of the same species, under different stresses. Particularly, it varies from ng/g to μg/g in different organs of the same plant (*Wang et al., 202*). In recent years, it has been reported that melatonin, as a stress regulator, can enhance a plant's tolerance to environmental stress, including drought (*Hosseini et al., 2021*), waterlogging (*Zhang et al., 2019b*), cold (*Li et al., 2018b*), salinity (*Li et al., 2019*), and heavy metals (*Yu et al., 2021*). *Huang et al. (2019)* found that exogenous melatonin reduced the downtrend of net photosynthetic rate and stomatal conductance in maize leaves. *Li et al. (2017)* noted that exogenous melatonin alleviated membrane damage in watermelon seedlings caused by stress-induced ROS burst. Several studies have shown that exogenous melatonin application enhances drought tolerance in *Malus domestica* (*Wang et al., 2013*) and *Davidia involucrata* (*Liu et al., 2021*). *Liu et al. (2015)* also found that 0.1 mM melatonin significantly alleviated membrane lipid peroxidation and enhanced drought tolerance in tomato seedlings. Earlier research has demonstrated melatonin's inability to scavenge $O_2^{\bullet-}$ and $H_2O_2$ directly; however, it assists in regulating ROS production and scavenges through improved antioxidant enzyme activities (*Huang et al., 2019*; *Li et al., 2017*). Furthermore, there is very little knowledge about the effect of exogenous melatonin on persimmon rootstock's response to drought stress.

Nitric oxide (NO), a redox signaling molecule, regulates steady ROS production through activated antioxidant enzymes in plants (*Nabi et al., 2019*). Melatonin, ROS, and NO are essential for a plant's response to abiotic stresses such as drought (*Gong et al., 2017*), cold (*Jahan et al., 2019*), heat, and salinity (*Arnao & Hernández-Ruiz, 2019*). However, the interaction between melatonin, ROS, and NO in *D. lotus*' response to drought remains unclear.

This study evaluated the effects of melatonin on *D. lotus* by analyzing the NO concentration, MDA, ROS concentration, and antioxidant enzymes and the key genes involved in the drought response.

## MATERIALS & METHODS

### Plant material and experiment design

The surface sterilized seeds of Diospyros lotus were immerged at 40 °C in sand for 2 days then stratified in sand for 20 days. After germination, seeds were planted in nursery seedling plates. After cultivation for 40 days under natural light and temperature conditions (day/night temperature, 28/20 °C; relative air humidity, 50–60%; photosynthetically active radiation, 500 $\mu$mol m$^{-2}$ s$^{-1}$), seedlings with similar size (7–8 leaves, about 12 cm tall) were selected and transferred to aerated Hoagland nutrient solution (*Hoagland & Arnon, 1950* Table S1). The Hoagland solution was refreshed every three days. After four weeks, plants with similar height and growth performance were divided into six groups (six plants in each group): Plants with similar height and growth performance from each species were divided into six groups (6 plants in each group): well-watered (CK), Hoagland solution containing 20% PEG-6000 (D); Hoagland solution containing 20% PEG-6000 and 10 $\mu$M melatonin (M1+D); Hoagland solution containing 20% PEG-6000 and 20 $\mu$M melatonin (M2+D); Hoagland solution containing 20% PEG-6000 and 50 $\mu$M melatonin (M3+D), Hoagland solution containing 20% PEG-6000 and 100 $\mu$M melatonin (M4+D). Root and the second leaves from the top were harvested after 2 d of treatments. Then, they were snap frozen in liquid nitrogen and stored at $-80$ °C for the following measurements.

### Determination of Chlorophyll fluorescence

Chlorophyll fluorescence was determined using a Chl-fluorescence Analyzer (Yaxin-1161G, Beijing Yaxinliyi Science and Technology Co., Ltd). The maximum efficiency of PSII photochemistry (Fv/Fm) were calculated after adaption in the dark for 30 min.

### Determination of superoxide, H$_2$O$_2$ and Malonaldehyde

The concentrations of malonaldehyde (MDA) in root and leaf tissues were recorded at 450, 532, and 600 nm using a spectrophotometer (TU-1810, Beijing Purkinje General Instrument Co., Ltd) as suggested by *Zhang et al. (2017)*.

Concentrations of the superoxide (O$_2^{\bullet-}$) and H$_2$O$_2$ in root and leaf tissues were determined spectrophotometrically at 530 and 410 nm, respectively, as suggested by *Zhou et al. (2017)*.
## Analysis of antioxidative enzyme activities

Fresh leaf and root samples (0.2 g) were ground in liquid nitrogen and homogenized with high-throughput tissue grinder equipment (Scientz-48L; Ningbo Scientz Biotechnology Co., Ltd). The soluble proteins in fresh plant tissues were extracted and quantified as described (*Zhang et al., 2017*). The activity of superoxide dismutase (SOD), catalase (CAT), guaiacol peroxidase (GPX), ascorbate peroxidase (APX) was determined as suggested by *Zhou et al. (2017)*, and glutathione reductase (GR) according to the method of *Edwards, Rawsthorne & Mullineaux (1990)*.

## Determination of nitric oxide content and nitrate reductase activity

The concentrations of nitric oxide were determined by using a A013-2-1 nitric oxide (NO) assay kit (Nanjing Jiancheng Bioengineering Institute, China). Nitrate reductase (NR) activity was determined as suggested by *Kaiser & Huber (1997)*.

## qRT-PCR analysis

Total RNA extraction and RT-PCR were carried out by the method of *Zhang et al. (2017)*. Total RNA of root was isolated and purified by using a plant RNA extraction kit (R6827, Omega Bio-Tek, Norcross, GA, USA). The quality and concentration of extracted RNA was measured by agarose gel electrophoresis and spectrophotometer analysis (NanoDrop 2000; Thermo Fisher Scientific Ltd., Waltham, MA, USA). qRT-PCR was performed according to 10 µl 2×SYBR Green Premix Ex Taq II (DRR820A; Takara, Dalian, China), 0.5 µl cDNA, and 0.2 µl primer, and then using a CFX96 Real Time system to test (CFX96, Bio-Rad, Hercules, CA, USA). The $\beta$-Actin was served as a reference gene. All primers used are listed in Table S2. The relative mRNA expression was calculated using the $2^{-\Delta\Delta CT}$ method (*Livaka & Schmittgen, 2001*).

## Statistical analysis

A one-way analysis of variance (ANOVA) was performed with PASW (IBM, Armonk, NY, USA). All data were analyzed via the Duncan method and were considered significant at $p < 0.05$. Graphs were plotted using Origin 9.1.0 (b215; OriginLab Corp., Northampton, MA, USA).

# RESULTS

## Effects of exogenous melatonin on $O_2^{\bullet-}$, $H_2O_2$, and malondialdehyde content

The MDA content was enhanced significantly over 1.29-fold in the roots of *D. lotus* under drought conditions compared to CK. Compared to drought, the MDA concentration decreased gradually (0.31–0.45 fold) after melatonin application. However, no significant differences among melatonin treatments were observed except for 100 µM.

The present study also measured the concentration of ROS, such as $O_2^{\bullet-}$ and $H_2O_2$, which is stimulated in plants under drought conditions. The study observed significantly increased concentrations of $O_2^{\bullet-}$ and $H_2O_2$ in the roots and leaves of *D. lotus* under drought compared to the control treatment (Fig. 1). In leaves, the lowest concentration of $O_2^{\bullet-}$ was observed under M1+D treatment, while no significant differences were found

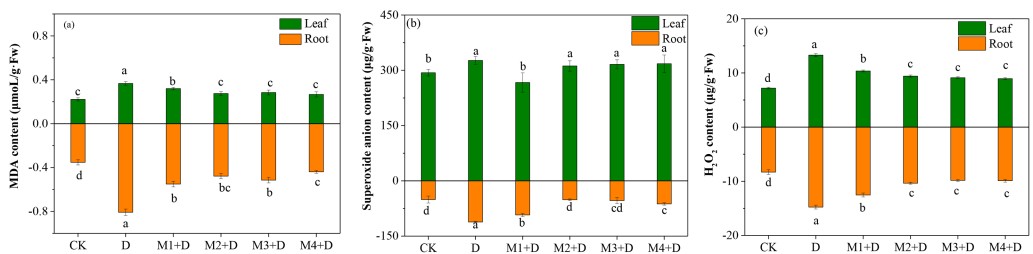

**Figure 1** **Effects of exogenous application of melatonin on MDA and ROS content in *D. lotus* roots and leaves under normal and drought stress conditions.**

among M2+D, M3+D, and M4+D. In roots, the $O_2^{\bullet-}$ concentrations were significantly suppressed under M2+D, M3+D, and M4+D compared to drought conditions. The $O_2^{\bullet-}$ concentrations in M1+D, M2+D, M3+D, and M4+D treatments were escalated by 82.24%, 1.14%, 5.43%, and 23.34%, respectively, compared to CK.

An increased level of melatonin application decreased the $H_2O_2$ contents both in the root and leave tissues of *D. lotus* under drought conditions, but the decrease was not significant. Compared with the control, the drought-treated *D. lotus* showed 64.26% and 90.45% higher $H_2O_2$ contents in roots and leaves, respectively, while relatively lower escalation by 50.92%, 24.76%, 18.39%, and 19.10% was observed in roots, 44.12%, 31.05, 26.96, and 24.51 in leaves under M1+D, M2+D, M3+D, and M4+D treatments respectively.

## Effects of exogenous melatonin on the maximum efficiency of photosystem II photochemistry ($F_v/F_m$)

The normalized ratio between variable fluorescence ($F_v$) and maximal fluorescence ($F_m$), $F_v/F_m$, is an effective measure of photosystem II (PSII) performance. The $F_v/F_m$ decreased significantly under drought conditions compared to CK (Fig. 2). Although $F_v/F_m$ elevated remarkably under exogenous melatonin application compared to drought stress, it decreased significantly compared to CK. It was also observed that 100 μM melatonin application was the most effective among other concentrations.

## Effects of exogenous melatonin on ROS production and scavenging enzymes

Nicotinamide adenine dinucleotide phosphatase (NAPDH) oxidase, also known as respiratory burst oxidase homolog (RBOH), provides localized ROS bursts to the environmental stress response (*Chapman et al., 2019*). Under drought treatment, the highest activity of NAPDH oxidase was found in roots and leaves of *D. lotus* (Fig. 3). However, no significant differences were found in *D. lotus* leaves under exogenous melatonin application compared to CK. However, the NAPDH oxidase activities increased significantly by 40.43% and 32.47% under M1+D and M2+D treatments, respectively, compared to CK.

Antioxidant enzymes play an essential role in ROS scavenging in plants exposed to various environmental stresses. The enzymatic antioxidants (SOD, POD, CAT, and APX) were examined in *D. lotus* under drought and melatonin treatments. Under drought

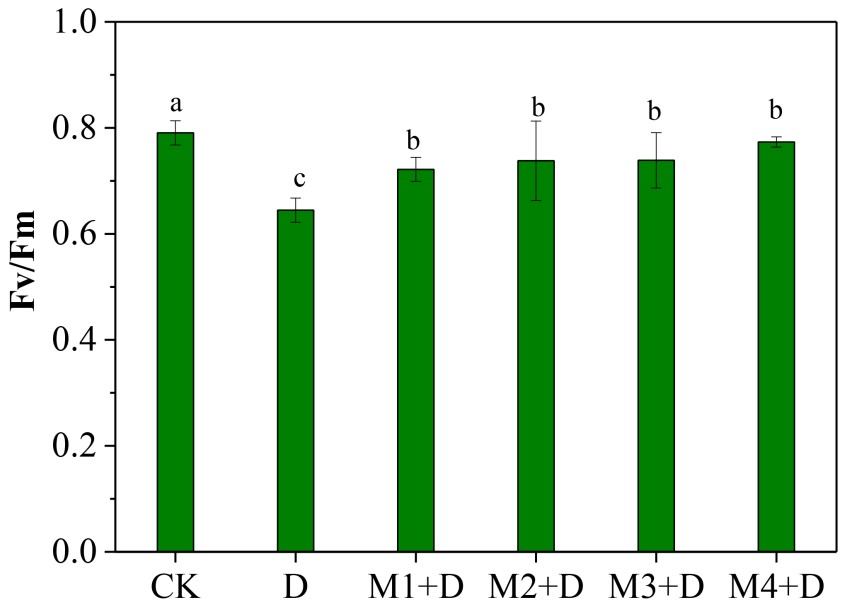

**Figure 2** Effects of exogenous application of melatonin on chlorophyll fluorescence in *D. lotus* leaves under normal and drought stress conditions.

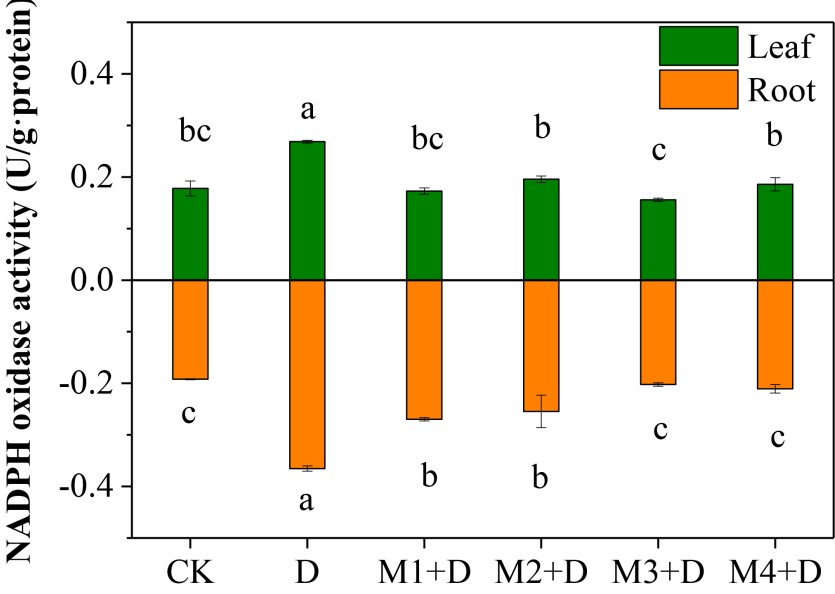

**Figure 3** Effects of exogenous application of melatonin on NAPDH oxidase activity in *D. lotus* roots and leaves under normal and drought stress conditions.

treatment, the SOD activities increased markedly in the roots by 10.37%, 1.03%, 1.28%, and 12.7% under M1+D, M2+D, M3+D, and M4+D treatments, respectively, compared to CK (Fig. 4A). The highest SOD activity among various melatonin treatments was observed

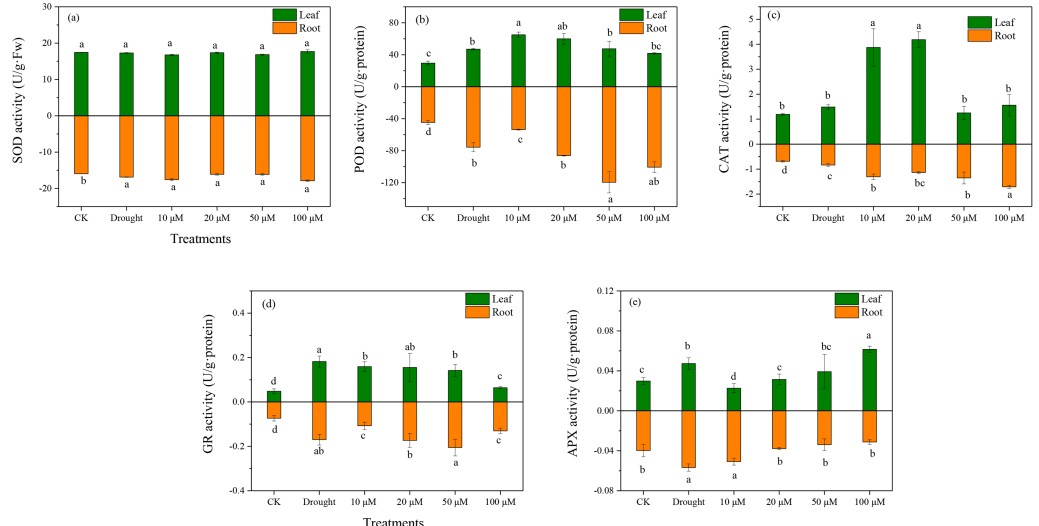

**Figure 4 Effects of exogenous application of melatonin on antioxidant enzyme activity in *D. lotus* roots and leaves under normal and drought stress conditions.**

under M4+D treatment, with no significant differences in *D. lotus* leaves. Melatonin enhanced POD activities in roots and leaves of *D. lotus* (Fig. 4B). In most cases, the CAT activities were higher in roots and leaves of *D. lotus* under drought and melatonin treatments (Fig. 4C). The CAT activities were elevated in the roots of melatonin treatments but were inhibited in the leaves exposed to 50 and 100 μM melatonin treatments compared to the drought treatment (Fig. 4C). The leaves of *D. lotus* under drought and melatonin treatments showed enhanced GR activities than the control treatment. However, the GR activities were the highest in *D. lotus* roots exposed to 50 μM melatonin compared to the control treatment (Fig. 4D). The leaves showed inhibited APX activities under 10 μM melatonin treatments compared to CK. In contrast, APX activities were elevated under 100 μM melatonin treatments (Fig. 4E). However, the highest APX activities were observed in *D. lotus* roots under drought conditions, while the lowest activities were observed in the roots under 100 μM melatonin application.

## Effects of exogenous melatonin on NO accumulation

The study examined the effect of different melatonin concentrations on NO accumulation in *D. lotus*. As shown in Fig. 5A, the leaves and roots of the drought-stressed plants released 143.71% and 12.72%, respectively, more NO than in the CK. Interestingly, melatonin application significantly changed the NO content both in the roots and leaves of the plants under drought conditions. In leaves, M1+D, M2+D, M3+D, M4+D increased the NO content by 71.49%, 70.22%, 165.82%, then decreased 17.42% compared to drought, respectively. However, the NO concentration in melatonin application leaves were higher in comparison with CK. In roots, increasing melatonin concentrations (0–50 μM) also

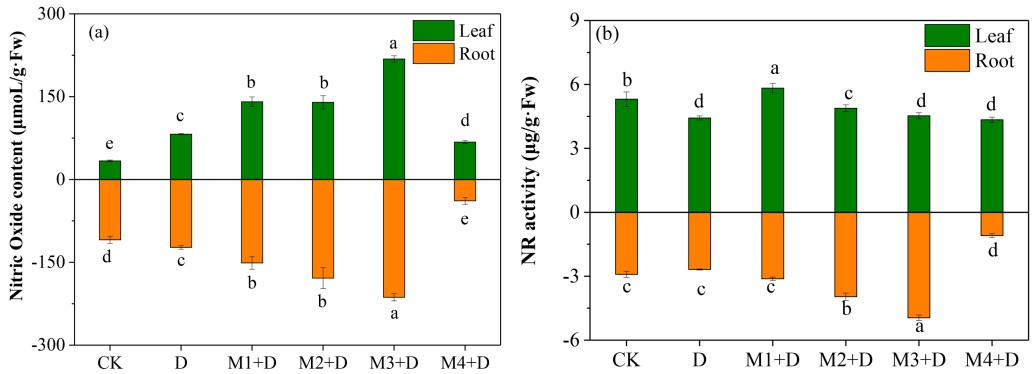

**Figure 5** Effects of exogenous application of melatonin on NO content and NR activity in *D. lotus* roots and leaves under normal and drought stress conditions.

increased the NO concentration. The NO concentration was the lowest in the roots exposed to 100 μM melatonin compared to CK.

NR is localized in the cytoplasm and participates in NO accumulation in response to the environmental stress (*Kaya et al., 2020*). The NR activity was inhibited by 16.60% and 7.82%, respectively, in the leaves and roots than the CK (Fig. 5B). Melatonin application significantly increased the NR activity in leaves in M1+D, M2+D, and M3+D treatments, reaching the lowest in M4+D treatment. In the roots, increased melatonin levels (10, 20, and 50 μM) significantly increased the NR activity (7.22%, 36.07%, and 70.15%, respectively) compared to CK. The NR activity was the lowest in the roots of *D. lotus* exposed to 100 μM melatonin.

## Effects of exogenous melatonin on key genes of ROS metabolism

*Diospyros lotus* under drought and melatonin treatments exhibited ROS generation and scavenging abilities, which is related to differential transcriptional regulation of key genes involved in ROS metabolism. Therefore, the study analyzed four genes' transcripts levels involved in ROS generation and scavenging in the roots (Fig. 6). RBOH plays a critical role in altering $H_2O_2$ production. Compared with the controls, drought stress resulted in an up-regulation of *DlRBOHA* in the root and leave tissues of *D. lotus*. The mRNA expression of *DlRBOHA* was the highest in leaves of *D. lotus* and was 0.4-fold up-regulated under drought and melatonin treatments compared to CK. The transcript levels of *DlRBOHA* in melatonin treatments were higher than CK, but they were down-regulated in the leaves of *D. lotus* (except for 50 μM) compared to the drought-stressed plants. The same trend was also observed in the roots.

The major antioxidant enzymes (SOD, CAT, and POD) contribute to ROS scavenging. The transcript levels of *DlSOD* were significantly up-regulated in the root and leave tissues of *D. lotus* under drought conditions, compared to the control. There were no significant differences among the various melatonin treatments except for 10 μM, but the transcript levels of *DlSOD* were still higher than CK in *D. lotus* leaves. Like in the leaves, the transcript levels of *DlSOD* in the roots of *D. lotus* were significantly up-regulated under melatonin

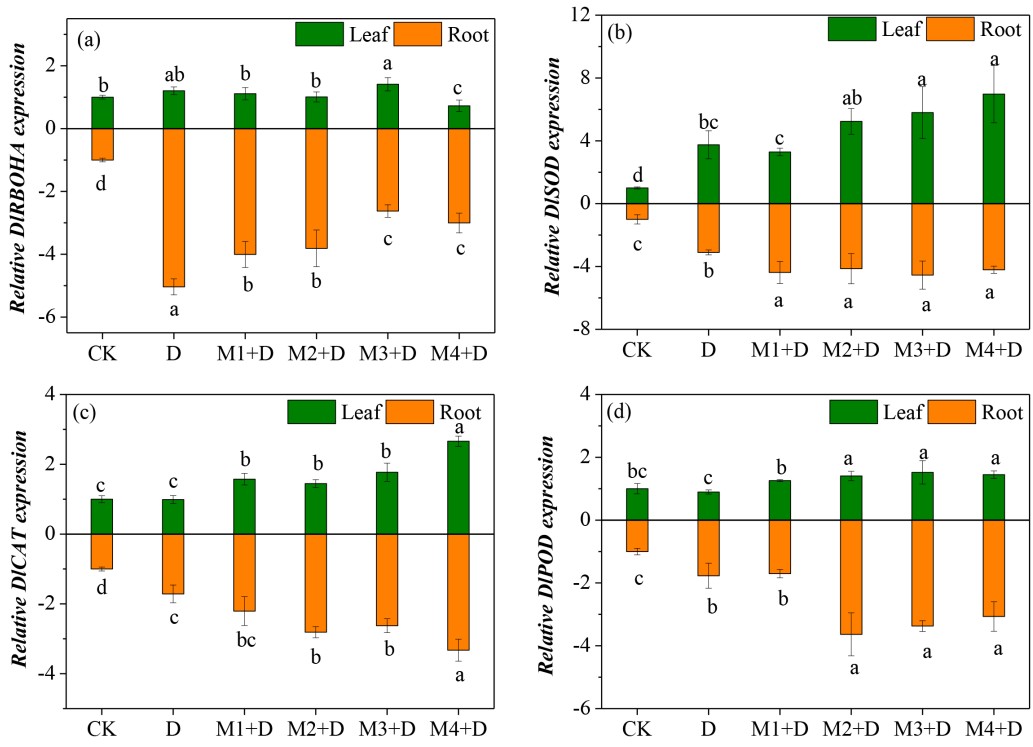

**Figure 6** (A–D) Effects of exogenous application of melatonin on the expression of key genes involved in ROS metabolism in *D. lotus* roots and leaves under normal and drought stress conditions.

treatments. Furthermore, the transcript levels of *DlCAT* in the leaves and roots of *D. lotus* were significantly up-regulated under melatonin treatments, with the highest being 166.01% (leaves) and 232.70% (roots) under 100 μM melatonin compared to CK. The *DlPOD* transcript levels also showed significant up-regulation under melatonin treatments. Thus, the results conclusively indicated that melatonin mitigates ROS damage by enhancing the gene expression for antioxidant enzyme activities in *D. lotus*.

## DISCUSSION

In recent years, extreme drought conditions in North China have limited fruit-tree development and production (*Yang et al., 2021*). *Huang et al. (2019)* reported tissue-specific responses to drought stress at the level of ROS, plant growth, and antioxidant system in maize seedlings. Drought stress inhibited the shoots more than the roots in soybean seedlings (*Du et al., 2020*). Our research observed increased accumulation of MDA, superoxidase, and $H_2O_2$ in *D. lotus* under the drought stress. Additionally, organ-specific responses were also observed (Fig. 1). Compared with leaves, the MDA and ROS concentrations were higher in the roots under drought conditions. The roots sense water deficit directly, therefore, maintaining an efficient ROS scavenging system could benefit drought tolerance.

A useful strategy for improving seedling abiotic stress tolerance is the use of phytohormones (*Jogawat et al., 2021*; *Liang et al., 2019*). The primary function of melatonin, along with its effects on the antioxidant enzyme activity, may be to maintain intracellular ROS homeostasis under the environmental stress (*Bidabadi, VanderWeide & Sabbatini, 2020*; *Liu et al., 2021*). Exogenous melatonin attenuated drought-induced cell damage in *Cynodon dactylon* (*Shi et al., 2014*) and *Solanum lycopersicum* (*Liu et al., 2015*). Ahmad et al. (2019b) revealed that foliar spray of melatonin alleviated oxidative damage in maize seedlings. Our results demonstrated that exogenous melatonin application suppressed the drought-induced ROS burst, enhancing the drought tolerance. Under drought stress, plants evolve tolerance mechanisms, such as enzymatic ROS scavenging system that inhibits ROS bursts, to maintain ROS homeostasis (*Abid et al., 2018*; *Imran et al., 2021*; *Li et al., 2018a*). Many studies have reported decreased activities of SOD, POD, CAT, and APX during drought stress (*Imran et al., 2021*); melatonin application significantly increased the activities of these enzymes, inhibiting ROS accumulation (*Shi et al., 2015*).

Exogenous melatonin alleviated ROS accumulation and drought-induced ROS bursts not only by enhancing the activities of antioxidant enzymes (SOD, POD, CAT, and APX) in *Fagopyrum tataricum* (*Hossain et al., 2020*), but also enhancing the mRNA expression of *SOD*, *CAT*, *APX*, and *POD* that scavenge excess ROS (*Sharma & Zheng, 2019*). It is found that the melatonin-treated *D. lotus* plant shows reduced MDA and ROS content, and this could be related to antioxidant enzyme activities such as SOD, CAT, GR, and APX (Fig. 4). Furthermore, the transcript levels of genes related to ROS metabolism, *i.e.*, *DlSOD*, *DlPOD*, *DlCAT*, *DlRBOHA* were up-regulated by exogenous melatonin. These results are consistent with those reported by *Altaf et al. (2022)*, who discovered that exogenous melatonin mitigated the antioxidant enzymes by regulating the gene expression for *SOD*, *CAT*, *APX*, *GR*, and *POD*.

Recent research found that exogenous melatonin could be transmitted from roots to leaves; exogenous melatonin root irrigation not only reduced ROS bursts in the roots but also improved photosynthesis in the leaves; and the melatonin concentration regulated its mitigation potential in drought-stressed plants (*Altaf et al., 2022*; *Campos et al., 2019*; *Imran et al., 2021*; *Liang et al., 2019*). *Imran et al. (2021)* showed root irrigation of melatonin to be more effective than foliar application; 100 μM melatonin application was more effective than other concentrations. Our research showed that drought conditions decreased chlorophyll fluorescence ability ($F_v/F_m$) in *D. lotus* leaves. However, exogenous melatonin application could partly relieve the decrease. Earlier research has shown that melatonin application increased $F_v/F_m$, the effective quantum yield of PSII, and the photochemical quenching in maize seedlings leaves under drought stress (*Huang et al., 2019*). This study observed 100 μM melatonin application to provide the best protection against drought stress and was more effective than other concentrations (Fig. 2).

This research demonstrated that exogenous melatonin elevated the endogenous NO concentration in *D. lotus* roots under drought conditions. *Imran et al. (2021)* also reported that 100 μM melatonin significantly enhanced NO content in *Glycine max* under drought stress. Nitric oxide is often considered a second signal for a plant's stress response (*Zhu et*

al., 2019). In heavy metal stress, melatonin may interfere with NO-mediated root cell cycle progression and the quiescent center cell homeostasis, alleviating *Arabidopsis* root growth inhibition (Zhang et al., 2019a). However, melatonin's interaction with NO in drought stress remains unclear. He & He (2020) indicated an interaction between melatonin and NO, with melatonin acting as an NO scavenger. The current study investigated the NO content and NO-related gene expressions under melatonin applications and observed that low concentrations of melatonin (0–50 μM) promoted NO concentration. However, high melatonin 100 μM) concentration inhibited NO concentration (Fig. 5). Furthermore, melatonin application probably induced the mRNA expression of NR.

Many studies demonstrated the dose-dependent response of melatonin in plants tolerant to salt (Ahmad et al., 2021a), cold (Chang et al., 2022), high temperature (Yu et al., 2022), and heavy metals (Nabaei & Amooaghaie, 2019). Ahmad et al. (2019) found that foliar application of 100 μM was more effective in improving maize seedling's drought tolerance than 50 and 75 μM melatonin application. Campos et al. (2019) reported that 500 μM melatonin reduced drought tolerance in *Coffea arabica*, while 300 μM melatonin application protected *Coffea arabica* against drought stress. In the current study, exogenous melatonin application alleviated oxidative damage in *D. lotus* leaves and roots, especially the 100-μM melatonin application, which suggests that under drought stress, exogenous melatonin treatment effectively protects the cell membrane against oxidative damage. However, the most effective does not necessarily mean the most economical. Our study results showed no significant differences between 20 and 100 μM melatonin applications, hence 20 μM melatonin could be an economical option for large-scale field use.

## CONCLUSIONS

Our study demonstrated that the melatonin-induced improvement in drought-stress tolerance in *D. lotus* was associated with an enhanced enzymatic ROS scavenging system. Melatonin application activated major antioxidant enzymes such as superoxide dismutase, catalase, peroxidase, glutathione reductase, and ascorbate peroxidase. Exogenous application of melatonin, especially at 100 μM concentration, significantly inhibited drought-induced damage. However, there were no significant differences between 20 and 100 μM melatonin applications, hence 20 μM melatonin application would be highly economical for large-scale use. Additionally, NO concentration was significantly higher in 10 and 50 μM melatonin treatments and lower in 100 μM melatonin treatment compared to the control. the possible interaction between melatonin and NO may play a crucial role in drought-stress tolerance in *D. lotus*.

### Abbreviations

| | |
|---|---|
| **ROS** | Reactive oxygen species |
| **SOD** | superoxide dismutase |
| **CAT** | catalase |
| **POD** | peroxidase |
| **APX** | ascorbate peroxidase |
| **GR** | glutathione reductase |

| NO | Nitric oxide |
|---|---|
| $H_2O_2$ | Hydrogen peroxide |
| Fv/Fm | Maximal quantum yield of PSII photochemistry |
| NR | Nitrate reductase |

### Funding

This work was funded by the Scientific and Technological Research Project of Henan Province (No. 202102110051 and 202102110054), and the Scientific Research Starting Foundation of Henan Institute of Science and Technology (No. 103010620001/003). The funders had no role in study design, data collection and analysis, decision to publish, or preparation of the manuscript.

### Grant Disclosures

The following grant information was disclosed by the authors:
Scientific and Technological Research Project of Henan Province: 202102110051, 202102110054.
Scientific Research Starting Foundation of Henan Institute of Science and Technology: 103010620001/003.

### Competing Interests

The authors declare there are no competing interests.

### Author Contributions

- Peng Zhang conceived and designed the experiments, performed the experiments, analyzed the data, prepared figures and/or tables, authored or reviewed drafts of the article, and approved the final draft.
- Yi Hu performed the experiments, analyzed the data, prepared figures and/or tables, and approved the final draft.
- Ruijin Zhou conceived and designed the experiments, performed the experiments, authored or reviewed drafts of the article, and approved the final draft.
- Xiaona Zhang analyzed the data, authored or reviewed drafts of the article, and approved the final draft.
- Huiling Hu conceived and designed the experiments, authored or reviewed drafts of the article, and approved the final draft.
- Dongmei Lang conceived and designed the experiments, authored or reviewed drafts of the article, and approved the final draft.

### Data Availability

The raw data is available at Zenodo: Zhangpeng. (2022). The antioxidant system response to drought-stressed Diospyros lotus treated with exogenous melatonin [Data set]. Zenodo. https://doi.org/10.5281/zenodo.6526608.

## Supplemental Information

Supplemental information for this article can be found online at http://dx.doi.org/10.7717/peerj.13936#supplemental-information.

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
