# Peer review of "The antioxidant system response to drought-stressed Diospyros lotus treated with exogenous melatonin"

_PeerJ, doi:10.7717/peerj.13936_

## Round 0.1 · original submission · Major Revisions

Dear authors, you need major modification of your MS. Please revise the MS according to the reviewers comments.

·

Basic reporting

This manuscript reported how exogenous melatonin application could mitigate drought stress in persimmon seedlings. The quality of the presentation is satisfactory, but the overall writing and English should be improved. This manuscript also reported valuable and noble findings that could be interesting to many readers. The introduction should be improved. The key research questions are not clear.
The figure X-axis notation should be revised. From 10um-100um melatonin treatment, it is not apparent whether the drought was applied or not. It seems that only melatonin was used. It should be revised to indicate the presence of both melatonin and drought.
L65: Revise the sentence. It is not logically clear.
L74-75: A generalized sentence does not represent why the whole paragraph was written. Even it is somehow a repetition of L57-58. Focus on more specific points.
L81-82: revise the objectives. The word 'oxidants' does not sounds well.

Experimental design

The manuscript is well-designed, but the authors failed to describe the experimental procedures in detail in the method section.
The central issue of this study is that the plants were exposed to treatment conditions for only two days. The authors must have explained why they only continued the experiment for two days in the manuscript. The objective should be revised. The experimental findings are adequate to support the objectives of this study.
L87: What was the 'seedling substrate.' Add details in the manuscript.
L89: Add a citation for Hoagland nutrient solution.
L90: On which day was the PEG added? Add in the manuscript.
How was the 20% PEG added? At one time or at several intervals. Add in details.
L91: 'from each species.' Not clear.
L95: only "2 d" after treatment samples were collected. Is it 2 days? Isn't it a very short time to observe the changes or draw a conclusion?
It is a major drawback of the manuscript. Authors must have explained: 'Why only two days after treatment?"
When was the melatonin applied? How and how many times was melatonin applied?

Validity of the findings

Although the authors observed gene expression of a few antioxidant enzymes, but no description was added related to these in the discussion section.
Why did the SOD activity not change due to the application of melatonin and drought stress?

Additional comments

L16-17: Logical connection between the first and second sentences is not good. Please revise.
L17: It should be better to mention 'PGR.'
L22: 'nitric oxide' should be abbreviated here. In L26, use the abbreviation only.
L25: The word 'activated' is not suitable for antioxidant enzymes. Revise it.
L26-27: The sentence regarding "NO" is irrelevant in the abstract section.
L29-30: Revise the sentence. It is already proved that MT has physiological roles in plants. Focus on your results and try to revise the conclusion based on the objectives.
L42: 'one of the main reasons for drought..' This phrase changes the meaning of the sentence in the wrong direction. Revise the sentence.
L49: Physiological or Biochemical or both?
L243: The logic of using 'Another....' is not apparent here. Did the authors use methods to mitigate drought stress in this paper?

Reviewer 2 ·

Basic reporting

The manuscript addresses the physiological responses elicited in drought stressed Diospyros lotus by application of melatonin.
The authors provide data on chlorophyll fluorescence, a good indicator of stress;
- metabolites associated with stress response MDA, superoxide, hydrogen peroxide, nitric oxide
and on antioxidant enzymes
The Introduction does provide background to the study. Some issues to be addressed.
1. Line 36 it is not apparent to an international reader that the CNY is referring to Chinese currency. Please write in full.
2. Line 40 seems there is unnecessary repetition …diminishes net photosynthesis and decreases photosynthetic capacity.
3. Statement in Line 42-43 is not correct. One of the main reasons for drought…….did the authors mean one of the main ‘effects of drought……’ ?
4. Line 46 ‘develop’ is not the best word, since plants already have these mechanisms in place, they simply ‘turn them on’ or rather express. Suggestion to re-write the statement.
5. Line 59 following: Please provide a bit more detail on endogenous melatonin in plants, when is it produced, in what tissues, physiological concentrations? It seems to be a fairly new growth regulator.

Experimental design

Research is original. There is not much information about melatonin., a fairly new plant growth regulator. Experiment was well replicated with 6 reps. Some important details are missing in the methods.
1. Line 85-86 Please clarify the purpose of the mentioned seed treatments.
2. Line 87 Please specify what the seedling substrate is composed of.
3. Line 89…….; transferred to aerated……
4. Measurement of chlorophyll fluorescence, which leaf was measured, what time of day were measurements were taken.
5. Determination of superoxide, H2O2, and malonaldehyde …..You mentioned using the spectrophotometer to quantify, How were the samples prepared ?
6. Throughout the paper, several sentences should make a paragraph. Avoid taking one sentence as a paragraph.
7. Line 108 grammatical error……samples were ground …..not grounded.
8. Line 132 grammatical error
9. All figure legends always appear at the bottom of the figure not above.

1. There is a mix up of figure 4 and Figure 5 and their legends. The legend for Fig 4 is on 5 and vice versa.
2. Line 241 error
3. Line 252-253 Grammatical error on ‘Many research has……’

Validity of the findings

The results are interesting. Melatonin is a fairly new plant regulator. This information helps to build up information on the mechanism of action for this plant growth factor.

The discussion could be improved to make it more of a discussion of your results rather than literature review. Grammatical errors need to be fixed throughout the manuscript.

Reviewer 3 ·

Basic reporting

The antioxidant system response to drought-stressed Diospyros lotus treated with exogenous melatonin by Zhang et al needs major revision before its acceptance
Gene expression data please provide in abstract that is missing
Line 45in introduction please remove 8-10 instead add reference there.
Add role of chlorophyll fluroscence under drought in introduction section
Add in Material and methods are seeds surface sterilized or not
Also add PAR and growth conditions in materials and methods
Provide Hoagland solution in Material and methods section
Line 241 please corrects it.
Discussion should be improved and few latest publications should be incorporated
Conclusions should be improved as per abstract

Experimental design

The antioxidant system response to drought-stressed Diospyros lotus treated with exogenous melatonin by Zhang et al needs major revision before its acceptance
Gene expression data please provide in abstract that is missing
Line 45in introduction please remove 8-10 instead add reference there.
Add role of chlorophyll fluroscence under drought in introduction section
Add in Material and methods are seeds surface sterilized or not
Also add PAR and growth conditions in materials and methods
Provide Hoagland solution in Material and methods section
Line 241 please corrects it.
Discussion should be improved and few latest publications should be incorporated
Conclusions should be improved as per abstract

Validity of the findings

The antioxidant system response to drought-stressed Diospyros lotus treated with exogenous melatonin by Zhang et al needs major revision before its acceptance
Gene expression data please provide in abstract that is missing
Line 45in introduction please remove 8-10 instead add reference there.
Add role of chlorophyll fluroscence under drought in introduction section
Add in Material and methods are seeds surface sterilized or not
Also add PAR and growth conditions in materials and methods
Provide Hoagland solution in Material and methods section
Line 241 please corrects it.
Discussion should be improved and few latest publications should be incorporated
Conclusions should be improved as per abstract

Annotated reviews are not available for download in order to protect the identity of reviewers who chose to remain anonymous.

---

## Round 0.2 · accepted · Accept

The revised version of the MS has been endorsed for publication by the reviewers.

As PeerJ is primarily an English language journal, perhaps the mention of currency at line 43 might be also expressed in $US.

Other suggested edits are also included below.

LINE NO: / BEFORE / AFTER / [COMMENTS]
LINE 96: / plate / plates / [.]
LINE 104: / contains / containing / [.]
LINE 105: / contains / containing / [.]
LINE 105: / contains / containing / [.]
LINE 106: / contains / containing / [.]
LINE 107: / contains / containing / [.]
LINE 122: / High-throughput / high-throughput / [.]
LINE 128: / by using / by using a / [.]
LINE 155: / overserved / observed / [.]

·

Basic reporting

Based on the comments of the reviewers, this manuscript has been substantially revised. This manuscript might be published.But I have one suggestion.
1. The details meaning of the various treatment abbreviations should be included in the figure legends.

Experimental design

These issues were thoroughly explained by the authors.

Validity of the findings

Authors carefully resolved these issues.

Additional comments

Authors carefully resolved these issues.

Reviewer 2 ·

Basic reporting

Revisions accepted

Experimental design

Revisions accepted

Validity of the findings

Revisions accepted

Additional comments

Revisions accepted

Reviewer 3 ·

Basic reporting

The manuscript is suitable for publications
line no 109 in Materials and methods, plant name should be in italics

Experimental design

The manuscript is suitable for publications

Validity of the findings

The manuscript is suitable for publications